# Interconnected Set of Enzymes Provide Lysine Biosynthetic Intermediates and Ornithine Derivatives as Key Precursors for the Biosynthesis of Bioactive Secondary Metabolites

**DOI:** 10.3390/antibiotics12010159

**Published:** 2023-01-12

**Authors:** Paloma Liras, Juan Francisco Martín

**Affiliations:** Area de Microbiología, Departamento de Biología Molecular, Universidad de León, 24071 León, Spain

**Keywords:** lysine, ornithine, α-aminoadipic acid, α-aminoadipate semialdehyde, saccharopine, pipecolic acid, siderophores

## Abstract

Bacteria, filamentous fungi, and plants synthesize thousands of secondary metabolites with important biological and pharmacological activities. The biosynthesis of these metabolites is performed by networks of complex enzymes such as non-ribosomal peptide synthetases, polyketide synthases, and terpenoid biosynthetic enzymes. The efficient production of these metabolites is dependent upon the supply of precursors that arise from primary metabolism. In the last decades, an impressive array of biosynthetic enzymes that provide specific precursors and intermediates leading to secondary metabolites biosynthesis has been reported. Suitable knowledge of the elaborated pathways that synthesize these precursors or intermediates is essential for advancing chemical biology and the production of natural or semisynthetic biological products. Two of the more prolific routes that provide key precursors in the biosynthesis of antitumor, immunosuppressant, antifungal, or antibacterial compounds are the lysine and ornithine pathways, which are involved in the biosynthesis of β-lactams and other non-ribosomal peptides, and bacterial and fungal siderophores. Detailed analysis of the molecular genetics and biochemistry of the enzyme system shows that they are formed by closely related components. Particularly the focus of this study is on molecular genetics and the enzymatic steps that lead to the formation of intermediates of the lysine pathway, such as α-aminoadipic acid, saccharopine, pipecolic acid, and related compounds, and of ornithine-derived molecules, such as N^5^-Acetyl-N^5^-Hydroxyornithine and N^5^-anhydromevalonyl-N^5^-hydroxyornithine, which are precursors of siderophores. We provide evidence that shows interesting functional relationships between the genes encoding the enzymes that synthesize these products. This information will contribute to a better understanding of the possibilities of advancing the industrial applications of synthetic biology.

## 1. Introduction

Thousands of secondary metabolites (also named specialized metabolites) are synthesized by plants, fungi, and bacteria [1]. These metabolites show an impressive variety of chemical structures and have relevant biological roles in nature. Many of them show important pharmacological activities and are widely used in human and animal medicine [2,3]. This enormous variety of secondary metabolites is derived from a few dozen intermediates of the primary metabolism [4]. Several families of secondary metabolites arise from the basic amino acid lysine and ornithine and from intermediates or derivatives of the biosynthetic pathway of these two amino acids. Recent advances in the protein chemistry, molecular genetics, and biochemistry of lysine- and ornithine-derived compounds provide the basis for a better understanding of the pathways that supply precursors and intermediates for the biosynthesis of bioactive metabolites.

L-Lysine is an important basic amino acid that is synthesized in plants, fungi, and bacteria but is essential in humans and other mammals, which are unable to synthesize it and need to acquire it in their diet. However, it is well established that lysine is catabolized in mammals by the reverse of the final steps of the α-aminoadipate pathway that converts lysine into saccharopine (N^6^-glutaryl-L-lysine) and α-aminoadipate semialdehyde into different tissues: semialdehyde in bovine and primate liver and human placenta.

L-lysine is formed by three distinct pathways in different living beings. Plants, algae, and bacteria synthesize lysine from aspartic acid by the diaminopimelate pathway, whereas yeasts and filamentous fungi (except some lower fungi) use the α-aminoadipic acid (α-AAA) pathway, which starts from homocitrate [5,6]. Both pathways are shown in Figure 1A,B. Remarkably, in the last decades, it has been found that some thermophilic eubacteria, such as *Thermus thermophilus* [7,8,9], and archaea, e.g., *Sulfolobus acidocaldarius* [10] and *Thermococcus kodakarensis* [11], synthesize lysine by a modified α-aminoadipate pathway rather than the classical bacterial diaminopimelate pathway [12,13]. In this eubacterium and in archaea, the first half of the lysine biosynthesis pathway, up to α-AAA, follows the same steps as the fungal lysine pathway; in the second half of the pathway, the α-aminoadipate is converted into lysine through five steps that are similar to the initial steps of the arginine pathway leading from glutamic acid to ornithine (Figure 1C) using the same enzymes of the arginine pathway.

Lysine and its derivatives are critically important in the biosynthesis of many different secondary metabolites with antibiotic, immunosuppressant, antifungal, antitumor, and neuroprotective activities, among others. Similarly, ornithine derivatives are key components in the biosynthesis of bacterial and fungal siderophores. Many genes involved in the biosynthesis of lysine- and ornithine-derived secondary metabolites have been discovered but there is a lack of information on the correlation of the early enzyme studies and the knowledge of the molecular mechanisms provided by the genome sequencing and other omics tools.

Lysine biosynthetic reactions result in the formation of important intermediates such as α-AAA and α-aminoadipate semialdehyde, involved in the biosynthesis of β-lactam antibiotics [14]; moreover, lysine catabolic reactions form lysine derivatives such as saccharopine, pipecolic acid, and α-aminoadipate semialdehyde (Figure 2). These intermediates are interconverted, as will be detailed below. Pipecolic acid is a precursor of the antifungal and antitumoral compounds swainsonine and slaframine and of the immunosuppressants rapamycin, tacrolimus, and immunomycin. In addition, the pipecolic acid derivative N-hydroxypipecolic acid serves as an inducer of plant resistance to infections by pathogens. Suitable knowledge of the biosynthesis and availability of these precursors and intermediates is essential to developing new strategies in the search for novel bioactive metabolites. The lysine biosynthesis pathway in bacteria is also involved in the formation of DL-diaminopimelate, required in cell wall formation, and therefore, this pathway has been the subject of intense interest in the search for new antibiotics active against cell wall formation or remodeling, e.g., novel β-lactams and β-lactamase inhibitors [15]. In addition, the fungal α-AAA pathway, which does not exist in mammals, has been investigated in search of novel antifungal agents [16].

In this article, we review the available information on the molecular genetics and biochemistry of the biosynthesis of different precursors of secondary metabolites (Table 1), including novel data obtained from bioinformatic studies. The focus of this article is the analysis of secondary metabolites precursors in well-known model filamentous fungi, such as *Aspergillus nidulans*, *Aspergillus fumigatus*, and *Penicillium chrysogenum*, and actinobacteria, including *Streptomyces coelicolor*, *Streptomyces clavuligerus*, and *Streptomyces pristinaespiralis*; this knowledge in model organisms is then compared with the information available in other filamentous fungi and bacteria. We describe first the molecular mechanisms related to the biosynthesis of lysine-derived precursors (Section 2, Section 3 and Section 4), and the second part of the article focuses on ornithine derivatives that are precursors of siderophores in bacteria and fungi (Section 5, Section 6 and Section 7). We provide evidence indicating the existence of converging pathways for the biosynthesis of α-AAA and α-aminoadipate semialdehyde in filamentous fungi and Gram-positive bacteria (particularly actinobacteria, such as *Streptomyces* or *Nocardia* species) [17].

## 2. Biosynthesis of α-Aminoadipic Acid in Filamentous Fungi and Bacteria

α-Aminoadipic acid is a six-carbon analogous of aspartic acid and glutamic acid, and it acts as a specific inhibitor of the N-methyl-D-aspartate subtype of glutamate receptors in humans. This finding has created interest in organic chemical research, peptide chemistry, and neuroscience applications of this compound [40,41]. In addition, the interest in its role in secondary metabolite formation arises from the fact that β-lactam antibiotics contain an α-AAA component in their structures.

### 2.1. Biosynthesis of a-AAA and Interconversion of a-AAA and Lysine. Several Different Pathways Converge on the Biosynthesis of α-AAA in Filamentous Fungi (Figure 3)

#### 2.1.1. α-AAA Is Formed in Fungi as an Intermediate of the Lysine Pathway

The lysine pathway (Figure 3A) starts with the condensation of α-ketoglutarate and acetyl-CoA to form homocitrate. This seven-carbon tricarboxylic acid is converted to homoisocitrate that subsequently is decarboxylated to form α-ketoadipate. This compound is then converted to α-AAA by a poorly known α-ketoadipate aminotransferase [18]. There are two α-ketoadipate aminotransferases in *Saccharomyces cerevisiae*, one of which has broad substrate specificity [18], uses both α-ketoadipate and kynurenine as substrates, and is similar to human kynurenine aminotransferase [42]; a second aminotransferase in yeasts uses preferentially α-ketoglutarate as a substrate to form glutamic acid and accept α-ketoadipate with very low efficiency. The next enzyme in the pathway is an α-AAA reductase, encoded by the *lys2* gene, that converts α-AAA into its semialdehyde. This gene has been cloned, and the enzyme is characterized in *S. cerevisiae*, *Candida albicans* [27,43], and filamentous fungi [19,20]. The α-aminoadipate reductase (Lys2) is a large enzyme, very similar to a mono-modular NRPS, and is modified by a phosphopantetheinyl transferase (PPT encoded by *lys5*) that converts the inactive apo-Lys2 in the active holo-Lys2. Proteins homologous to Lys5 were also found in *A. nidulans* [44,45] and *P. chrysogenum* [46]. The Lys5 phosphopantetheinyl transferase of *P. chrysogenum* has 412 amino acids and belongs to the Sfp-type of PPTs [47]. This PPT is required for both lysine biosynthesis and penicillin production in *P. chrysogenum* since it also modifies the first enzyme of the penicillin pathway, (δ(L-α-aminoadipy-L-cysteinyl-D-valine)) synthetase; however, this PPT is different from the PPT required for fatty acid biosynthesis [46].

The last two steps of the fungal lysine pathway, carried out by the saccharopine reductase and saccharopine dehydrogenase, are reversible; these enzymes form first saccharopine from α-aminoadipate semialdehyde by condensation with glutamic acid, and finally lysine, by splitting the saccharopine into lysine and α-ketoglutarate. In a concerted action, these two enzymes convert α-aminoadipate semialdehyde to lysine by transfer of the amino group of glutamic acid to the C-6 carbon of α-aminoadipate semialdehyde. The enzymes of the lysine pathway up to the formation of α-AAA acid are located in mitochondria; the first enzyme, the homocitrate synthase, has mitochondrial and cytosolic isoenzymes derived from the same gene [48,49]. Electron microscopy studies of GFP-labeled homocitrate synthase demonstrated that most of the homocitrate synthase is located in the cytosol [50].

#### 2.1.2. Five Genes Encode Enzymes Similar to the Saccharopine Reductase and Saccharopine Dehydrogenase in Filamentous Fungi: A Possible Role for These Multiple Enzymes

In yeasts, two enzymes are well known to catalyze the conversion of α-aminoadipate semialdehyde to saccharopine and then to lysine; they are named (1) saccharopine reductase (saccharopine forming or glutamate forming in the reverse reaction) encoded by *Lys9* and, (2) saccharopine dehydrogenase (lysine forming) encoded by *Lys1* (reviewed by [5]). There is some confusion in the databases regarding the designation of these enzymes.

Remarkably, in some filamentous fungi, e.g., *P. chrysogenum*, there are up to five genes encoding these enzymes. The best known of these proteins is the saccharopine reductase encoded by *lys7.* This enzyme has been characterized in *P. chrysogenum*, including gene disruption, complementation, enzyme characteristics [22], and regulation studies [23]. This protein has 449 amino acids and corresponds to the CAC7475 (KZN5762) database entries. The *P. chrysogenum* saccharopine reductase is highly similar to the *Magnaporta grisea* homologous enzyme (66.6%) that has been crystalized [24,51]; the encoding gene, containing 10 introns, is located in chromosome III of *P. chrysogenum* P2niaD [22,52]. An orthologous gene occurs in the genome sequences of most filamentous fungi. A second putative reductase (45% identical to *lys7*), of 425 amino acids corresponds to KZN91882 and is restricted to some fungal species (e.g., *Penicillium nordicum*, 93% identity to KZN91882; *A. fumigatus* 76% and *Acremonium chrysogenum* 47%). This protein has an NAD^+^ binding domain in its amino terminal region (amino acids 7 to 122) as occurs in Lys7. The available information suggests that this enzyme is a duplicated saccharopine reductase in *P. chrysogenum*, although there are no enzymatic studies to confirm this hypothesis.

The saccharopine dehydrogenase in fungi has been identified by comparison with the well-known LYS1 protein of *S. cerevisiae.* A Blast search of the *P. chrysogenum* genome highlights a single protein corresponding to KZN9173 [53]. The gene encoding this protein is located in *P. chrysogenum* P2niaD chromosome I, encodes a 375 amino acids protein, and is ubiquitously present in all filamentous fungi.

Two other genes encoding enzymes annotated as saccharopine dehydrogenases are associated with the penicillin gene cluster. A total of 1 of 451 amino acids corresponds to ABA70586 (KZN89675) and is encoded by a gene located within the penicillin gene cluster, closely linked to *penDE*, which encodes the isopenicillin N acyltransferase [54,55]. The ABA70586 encoding gene is only present in the penicillin gene cluster of some penicillin producers, such as *P. chrysogenum*/*Penicillium rubens* strains, and in *Aspergillus oryzae*, which also produces penicillin. The presence of this highly expressed saccharopine dehydrogenase-like encoding gene in the penicillin gene cluster is of special interest since it is likely to enhance the supply of α-AAA precursor for penicillin biosynthesis [54]. Noteworthy, other β-lactam producers, such as *Penicillium nalgiovense*, *Penicillium griseofulvum*, *A. nidulans*, *A. chrysogenum*, *Emerocillopsis atlantica*, or *Trychophyton benhamanie*, lack in their penicillin gene clusters a gene similar to ABA70586. A gene encoding a protein homologous to ABA70586 occurs in *Aspergillus udawae* (73% identity) and *Aspergillus lentulus* but not in other *Aspergillus* species. Similar proteins also exist in *Acinetobacter baumanii* (45% identity) and *Streptomyces* and *Mycobacterium* species (around 40% identity), suggesting that this gene may be of bacterial origin [54,56].

The fifth related gene, encoding KZN89636, is located 39 kb downstream of the *penDE* gene in chromosome I in *P. chrysogenum* Wis 54-1255 [57] or chromosome II in *P. chrysogenum* P2niaD [52]. The KZN89636 and ABA70586 enzymes are related among themselves (46% identity, 63% similarity), both containing the NAD^+^ binding motif in their amino terminal regions and occurring only in some species of filamentous fungi.

In summary, it seems that *P. chrysogenum*, but not other penicillin producers, has acquired additional sacharopine dehydrogenase-like genes that are likely to influence the supply of precursors for penicillin biosynthesis.

### 2.2. Formation of α-AAA by Catabolism of Lysine

In addition to the above-described formation of α-AAA from homocitrate, α-AAA is also formed by catabolism of lysine through at least three different pathways: (a) it may be formed by reversal of the two last steps of the biosynthetic pathway that converts the first lysine into saccharopine, later into α-aminoadipate semialdehyde, and finally in α-aminoadipic acid; (b) by a lysine-6-aminotransferase (LAT), which converts lysine directly into α-aminoadipic semialdehyde (see Figure 3B); and (c) by a third pathway catalyzed by a saccharopine oxidase that directly forms α-aminoadipate semialdehyde from saccharopine.

#### 2.2.1. LAT-Dependent Biosynthesis of α-AAA in Filamentous Fungi and Bacteria

In *P. chrysogenum*, the enzyme lysine-6-aminotransferase forms α-aminoadipate semialdehyde from lysine. Characterization of *P. chrysogenum* LAT was approached by purifying the enzyme to homogeneity [58,59] and sequencing the N-terminal end of the pure protein (MATRG(S)(F)(S)HY). After the *P. chrysogenum* genome was sequenced [53], it was observed that this sequence corresponds to the protein KZN92070, unequivocally identifying the gene encoding the LAT enzyme in fungi. The LAT protein has 451 amino acids and uses as substrates L-lysine, L-ornithine, and with lower efficiency, N-acetyl-L-lysine as amino-group donors, thus the enzyme also received the name omega-aminotransferase because it deaminates the distal amino group of these substrates as amino group acceptors use mainly 2-ketoglutarate, 2-ketoadipate, and, to a lesser extent, pyruvate.

As indicated above, Gram-positive bacteria do not use the α-aminoadipate biosynthesis pathway; however, β-lactam-producing actinobacteria, such as *Nocardia lactamdurans*, *S. clavuligerus*, and *Streptomyces cattleya*, and some *Flavobaterium* species, contain a lysine-6-aminotransferase encoded by the *lat* gene located in the β-lactam gene cluster [21,60,61]. The LAT enzyme converts lysine into α-aminoadipate semialdehyde, a reaction identical to that of the omega-aminotransferase of filamentous fungi. LAT proteins, in both bacteria and fungi, have about 450 amino acids and are pyridoxal-phosphate-dependent enzymes, but their amino acid sequences have only 26% amino acid identity (42% similarity), suggesting an evolutionary functional convergent adaptation of distinct progenitor proteins to the lysine substrate. This enzyme is present in several actinobacteria but is not common in other groups of bacteria.

In bacteria, the α-aminoadipate semialdehyde is converted into α-aminoadipate by the piperideine-6-carboxylate dehydrogenase (PCD) that in *Streptomyces* producers of cephamycin is encoded by a gene (*pcd*) located in the β-lactam gene cluster [21,62,63]. *Flavobacterium lutescens* contain genes for LAT and for PCD [64,65], and it is known that *Flavobacterium lactamgenus* produces cephamycin-like antibiotics [66]. The LAT reaction product, α-aminoadipate semialdehyde, is highly reactive and toxic and is maintained intracellularly after dehydration in its cyclic form, piperidein-6-carboxylate (P6C). The preferent use of either α-aminoadipate semialdehyde or its cyclic form, P6C, as a substrate of PCD remains unclear. The *S. clavuligerus* PCD has been crystalized recently in its apo-form and also complexed with NADP, α-aminoadipic acid, or the P6C analogous picolinic acid, which contains a six-membered ring [67]. The structure of this protein is very similar to the human aldehyde dehydrogenase 7 (ALD7-1) and consists of three typical aldehyde reductase domains for NADP binding, substrate recognition, and dimerization. The binding site of the substrate analogous to picolinic acid was found to overlap with the substrate binding pocket, suggesting that the enzyme may recognize the substrate in its cyclic form (P6C).

Whereas in bacteria, the conversion of the α-aminoadipate semialdehyde into α-AAA has been shown to be catalyzed by the PCD enzyme, in fungi, it was believed that it is performed by reversal of the reaction carried out by the α-aminoadipate reductase, encoded by the *lys2* gene in the lysine biosynthesis pathway. However, LYS2 is a mono-modular NRPS that contains a reductase domain in the carboxyterminal region [19,68] (see Figure 1), and it is unclear if this reaction can be efficiently reversed.

Alternatively, the α-aminoadipate semialdehyde may be converted to α-AAA in fungi by an enzyme similar to the bacterial PCD. A search of the *P. chrysogenum* genome reveals four putative aldehyde reductases (KZN91999, KZN89995, KZN90899, KZN86644) with 30% to 32% identity to *S. clavuligerus* PCD (CAA04969) and a similar number of amino acids. However, no studies are available on these *P. chrysogenum* proteins, and it is unclear whether they recognize α-aminoadipate semialdehyde/P6C as a substrate.

#### 2.2.2. Lysine 6-Dehydrogenase-Dependent Biosynthesis of α-AAA in Bacteria

Additionally, in some bacteria, there is a gene encoding a lysine-6-dehydrogenase that performs a C-6 deaminase reaction similar to that catalyzed by LAT, but in contrast to LAT, the lysine-6-dehydrogenase is not a pyridoxal phosphate-dependent aminotransferase. The lysine-6-dehydrogenase has been well characterized by Misono et al. [69,70] in *Agrobacterium tumefaciens* (Figure 3B). This enzyme uses lysine and, with lower efficiency, S(β-aminoethyl)-L-cysteine as substrates and NAD^+^ as a cofactor. The lysine dehydrogenase is a dimer of 78 kDa, and its enzyme activity increases upon preincubation with lysine, which causes the formation of a tetramer. Using tritium-labeled substrates, it was found that the mechanism of action of the lysine dehydrogenase is the abstraction of the e-pro-R hydrogen at C-6 in the L-lysine.

Recently, a completely different mechanism of deamination of lysine at C-6 has been reported in some marine bacteria; detailed studies in *Marinomonas mediterranea* revealed a new enzyme responsible for this mechanism. The enzyme that performs this novel reaction is an L-lysine oxidase that uses a rare prosthetic group (CTquinone) formed by an internal post-translational cyclization of adjacent cysteine and tryptophan amino acids in the protein [71,72]. In summary, there are several pathways for converting lysine to α-aminoadipate semialdehyde and then to α-aminoadipate, some of which are parallel in bacteria and filamentous fungi.

## 3. Pipecolic Acid Biosynthesis in Fungi, Plants, and Bacteria

Pipecolic acid (also named nor-proline) is a six-carbon cyclic amino acid structurally similar to proline (Figure 2). This non-proteinogenic amino acid is a key intermediate in the biosynthesis of a large number of metabolites, e.g., alkaloids with antitumoral activity such as swainsonine and slaframine produced by the fungi *Slafractonia leguminicola*, *Metarhizium anisopliae* [73,74,75], and *Undifilum oxytropis*, an endophytic fungus that is toxic to grazing animals [76]. Pipecolic acid in bacteria is a precursor of pristinamycin I [25], friulimicin [77], meridamycin [78], rapamycin [79], tacrolimus [80], nocardiospin [81], and tubulysin B [82], among others. In fact, the presence of a pipecolic acid unit is essential for the bioactivity of immunosuppressants rapamycin and tacrolimus [83]. Pipecolic acid is also widely used in the chemical synthesis of a variety of drugs with interesting pharmacological activities [41].

### 3.1. Biosynthetic Routes and Enzymes Involved in the Formation of Pipecolic Acid in Fungi: Interconversion of Lysine and Pipecolic Acid

Pipecolic acid may be formed from lysine or from α-AAA (Figure 4). We address here first the formation of pipecolic acid from lysine. The enzymes and genes involved are shown in Appendix A. There are two different pathways for the biosynthesis of pipecolate from lysine in fungi. In both pathways, lysine is converted into saccharopine and then proceeds through the L-α-aminoadipate-6-semialdehyde (P6C) intermediate to pipecolic acid (Figure 4). However, there are differences in the enzymes converting saccharopine into P6C.

In one pathway, lysine is converted into saccharopine by reversal of the last step of lysine biosynthesis, and saccharopine is oxidized into P6C, releasing glutamate by the reverse saccharopine reductase [75]. A second alternative pathway in *S. leguminicola* contains a specific enzyme that converts saccharopine to P6C [76]. This enzyme, named saccharopine oxidase, was identified as a flavoenzyme of 45 kDa that requires oxygen and forms P6C, glutamate, and H_2_O_2_. The saccharopine oxidase is located in peroxisomes; this observation is important because many reactions involved in secondary metabolite formation take place in peroxisomes [84,85]. In the following step, the α-aminoadipate semialdehyde is reduced to form pipecolic acid by an NADP-dependent oxidoreductase; the same conversion is performed in vivo by a pyrroline-5-carboxylate reductase in an *Escherichia coli* transformant strain carrying the *lat* gene [86] indicating that the pipecolate oxidoreductase is functionally identical to pyrroline-5-carboxylate reductase of the proline pathway.

As indicated above, pipecolic acid is also formed from α-AAA, and it may be interconverted to lysine [22,23]. Using mutants blocked in the saccharopine reductase (*lys7*) or in the α-aminoadipate reductase (*lys2*), Naranjo et al. [22,23] demonstrated that pipecolic acid biosynthesis is greatly enhanced by the addition of α-AAA to a mutant disrupted in the saccharopine reductase, which is unable to convert saccharopine into P6C (and vice versa), whereas there is the very low formation of pipecolic acid in the *lys2* mutant disrupted in the α-aminoadipate reductase (Figure 4). These results strongly suggest that pipecolic acid is also formed from α-AAA through P6C, confirming the early evidence of Aspen and Meister [87].

### 3.2. Biosynthesis of Pipecolic Acid in Plants

In plants, pipecolic acid is a ubiquitous metabolite that is a precursor of the biosynthesis of many secondary metabolites. In addition, pipecolic acid regulates the resistance and immunity to bacterial infections in plants. This phenomenon, termed systemic acquired resistance (SAR) [88], results in a complete reprogramming of the metabolism in plants [89]. One of the key metabolites involved in genetic reprogramming after SAR is pipecolic acid [90,91,92,93]. The biosynthesis of pipecolic acid in plants requires two enzyme activities, as occurs in fungi, an aminotransferase that includes a cyclization and a reductase, but the intermediate, dehydropipecolic acid, is different from P6C [94] (Figure 4). In *Arabidopsis thaliana*, the aminotransferase ALD-1 transfers the amino group of lysine at C-2 to a ketoacid acceptor. The reaction product is enaminic-2,3-dehydropipecolic acid. Mutants disrupted in the ALD-1 gene are deficient in the formation of enaminic 2,3-dehydropipecolic acid and lack immunity to bacterial infections. These mutants are still able to convert 2,3-dehydropipecolic to pipecolic acid, confirming the involvement of an additional reductase activity. In *A. thaliana*, the reductase, functionally similar to the fungal pipecolate oxidoreductase, is named SARD4 and converts in vitro 2,3 dehydropipecolic acid into pipecolic acid [94].

Recently, the inducer molecule of systemic acquired resistance in *A. thaliana* has been identified as N-hydroxypipecolic acid [93]. This pipecolic acid derivative is synthesized by a FAD-dependent pipecolate monooxygenase (FMO1). The N-hydroxypipecolic acid acts as an inducer of a set of immune genes that increase cell resistance to microbial infections.

### 3.3. One Step Biosynthesis of Pipecolic Acid in Bacteria: Lysine Cyclodeaminases

In bacteria, pipecolic acid is a precursor of several antibiotics, immunosuppressants [76,94,95], and the antitumor agent VX710 [96] (Table 1). Several bacteria contain cyclodeaminases that directly catalyze the conversion of lysine into pipecolic acid or ornithine into proline.

Early studies showed that the biosynthesis of pipecolic by *S. hygroscopicus*, producer of rapamycin, is catalyzed by the *rapL* encoded lysine cyclodeaminase [26,80]; a similar enzyme encoded by *fkbL* is involved in tacrolimus (FK506) biosynthesis in *Steptomyces tsukubaensis* [97,98].

Remarkably, the bacterial cyclodeaminases are able to convert lysine into pipecolic acid in a single step [80,84]. This fact has created a great interest in the simple industrial enzymatic production of pipecolic acid as a head molecule for the chemical synthesis of many pharmaceutical drugs [99,100]. Early studies [101] using ^15^N-labeled lysine demonstrated that its amino group at carbon C-2 is removed, and the amino group of pipecolic acid derives from the lysine amino group at C-6. The mechanism of action of the cyclodeaminases was first studied in the ornithine cyclodeaminase that converts ornithine into proline [102]. These studies indicate that the cofactor NAD^+^ plays an important role in the mechanism of removal of the amino group. Additional research has focused on the molecular characterization of *S. pristinaespiralis* lysine cyclodeaminase [103,104,105,106]. This enzyme recognizes preferentially lysine but also accepts ornithine and 2,4-diaminobutiric acid as substrates. Crystallographic studies of this enzyme were performed with the enzyme bound either to NAD^+^, NAD^+^, and pipecolic acid or NAD^+^ and 2,4 diaminobutyric acid [104]. Characterization of the substrate pocket indicates that amino acid Asp**^236^** plays a key role in the binding of the lysine substrate. The mechanism of the reaction is rather unusual since NAD^+^ is converted into NADH and then reverted back to NAD^+^, which allows the release of the six-membered pipecolic acid and NAD+ [104,107].

## 4. 4-Oxopipecolic Acid and 3-Hydroxypicolinic Acid Precursors Related to Pipecolic Acid

4-oxopipecolic acid and 3-hydroxypicolinic acid are precursors of pristinamycin I. Early studies by Thibaud et al. [108] indicated that three NRPSs, named SnbA, SnbC, and SnbDE, are involved in pristinamycin I biosynthesis. The encoded proteins were purified, and the N-terminal end and internal peptides of SnbA were sequenced in order to clone the gene [28,109]. The first protein, SnbA, contains a loading module for 3-hydroxypicolinic acid (HPA), the starter amino acid in pristinamycin I biosynthesis. The second gene, *snbC*, encodes a bimodular NRPS that activates L-threonine and L-aminobutyric acid. The third gene, *snbDE*, encodes an NRPS containing four modules that activate L-proline, 4 N,N-dimethylamino-L-phenylacetic acid, 4-oxopipecolic acid, and L-phenylglycine [109,110]. Here, we focus this section on the biosynthesis of the lysine-derived 4-oxo-pipecolic acid and 3-hidroxypicolinic. The enzymes and genes involved in the biosynthesis of these molecules are shown in Appendix A.

### 4.1. Biosynthesis of 4-Oxo-L-Pipecolic Acid

The 4-oxo-L-pipecolic acid of *S. pristinaespiralis* is synthesized by two enzymes named PipA and SnbF, encoded by genes located in the pristinamycin I gene cluster [25]. The reaction mechanism of PipA, a lysine cyclodeaminase, has been elucidated [103,105]. PipA is homodimeric, and each monomer contains two functional domains, one to recognize lysine as substrate and the other containing a Rossman fold to interact with NADH. Based on the crystallographic studies, a model for the enzymatic reaction converting lysine in pipecolic acid has been proposed [102]. Pipecolic acid is then oxidized to 4-oxo-pipecolic acid by the P450 monooxygenase encoded by *snbF*; however, the substrate specificity of this monooxygenase has not been established so far. A gene (*visD*), homologous to *snbF*, is present in the virginiamycin S gene cluster. The protein encoded by *visD* is similar to the P450 cytochrome oxygenases that introduce hydroxyl groups in the macrolides erythromycin C (EryK) and tylosin (TylI) involved in the biosynthesis of these antibiotics in *Saccharopolyspora erythraea* and *Streptomyces fradiae*, respectively [29,111]. All these proteins have an oxygen-binding motif. Although the loading of the 4-oxo-pipecolic acid is performed by the third module of the synthetase SnbDE, there is no clear evidence of whether the pipecolic acid is oxidized to 4-oxo-pipecolic acid before or after binding to the NRPS.

### 4.2. Biosynthesis of 3-Hydroxypicolinic Acid

The biosynthesis of 3-hydroxypicolinic acid (Hpa), the starter unit of virginiamycin S and pristinamycin I, has been studied in detail in *Streptomyces virginiae*. Initially, Reed et al. [112] showed that 3-hydroxypicolinic acid derives from lysine, and later, an enzyme encoded by *visA* was proposed to be involved in the biosynthesis of this unit [29]. In *S. virginiae*, this gene encodes VisA, a pyridoxal-phosphate-dependent aminotransferase of 419 amino acids with dehydrase activity [29]. A homologous gene named *hpa* is present in the pristinamycin I gene cluster [25]. The HpaA protein has the characteristics of a lysine-2-aminotransferase that forms 2-keto-6-aminocaproic acid, which is then cyclized, forming piperideine-2-carboxylate and later 3-hydroxypicolinic acid. The HpaA and VisA aminotransferases share 66% identity, and a lysine residue in the active center (K^190^ in VisA, K^193^ in HpaA) has been proposed to be the pyridoxal-phosphate-binding site.

## 5. Cadaverine and Putrescine Precursors of Desferrioxamine-Type Siderophores

The key characteristic of siderophores is their extremely high affinity for iron. Siderophores play a pivotal role in cellular metabolism since iron is an essential component that participates in electron transport systems, respiratory chains, iron-sulfur clusters, and many redox reactions. However, if the iron concentration in the cells is abnormally high, it becomes toxic because iron reacts with oxygen by the Fenton reaction, producing highly reactive oxygen species. Some bacterial siderophores, such as desferrioxamines, are used in the medical treatment of high levels of iron in humans. Several siderophores have antibacterial activity, and they may be used in combination with antibiotics in the fight again pathogenic bacteria [113].

### 5.1. Biosynthesis of Desferrioxamines from Cadaverine

Desferrioxamines are one of the more important siderophores produced by many different species of *Streptomyces* and other bacteria. Desferrioxamines are formed by the repeated condensation of acylated N^6^-hydroxycadaverine linked by amide bonds; they include linear forms (desferrioxamine B) and cyclic molecules (e.g., desferrioxamine E). In the case of linear desferrioxamines, the structure is acylated by short- (acetyl) or medium-chain fatty acid units [114] (Figure 5). A gene (*desA)* involved in desferrioxamine biosynthesis was first observed in *Streptomyces pilosus*, a strain used for the industrial production of desferrioxamines for medical uses [115]. Later, the complete gene cluster *desABCD* was found in *S. coelicolor* [32,116]. Deletion of the *desA* or *desD* genes prevents desferrioxamine biosynthesis, providing solid evidence for the implication of the *desABCD* cluster in desferrioxamine biosynthesis [30,32]. Bioinformatic analysis [32] and lysine decarboxylase assays [117] allowed us to propose a role for the enzymes encoded by the four genes of the *desABCD* cluster. The first gene, *desA*, encodes a pyridoxal phosphate-dependent lysine decarboxylase that converts lysine into cadaverine, which is the basic building block of desferrioxamines [30,115]. This enzyme belongs to the family of 2,4-diaminobutyrate decarboxylases [117]. The second gene, *desB*, encodes an enzyme similar to flavin-dependent N-oxygenases [36,118] that seems to convert cadaverine or putrescine to their N-hydroxy derivatives. The third gene in the cluster, *desC*, encodes a wide-spectrum acyltransferase that has been proposed to catalyze the transfer of the succinyl group from succinyl-CoA to the amino group of N^6^-hydroxycadaverine, giving the N^6^-succinyl-N^6^- cadaverine intermediate of desferrioxamines. The last gene, *desD*, encodes an enzyme named desferrioxamine synthetase that catalyzes the final polymerization of three N^6^-succinyl-N^6^-hydroxycadaverine units and their cyclization to form desferrioxamine E (Figure 5) that is the major desferrioxamine produced by *S. coelicolor* [32]. The N-hydroxyl group of three N^6^-succinyl-N^6^-hydroxycadaverine molecules, together with the adjacent carbonyl group of each N-succinyl chain, form the N-hydroxamate structure that binds iron atoms with high affinity (Figure 5).

Several desferrioxamines differ in their acyl side chain since DesC is able to form N^6^-hydroxycadaverine acylated with different acyl groups, including acetyl, succinyl, or miristoyl chains [119]. In contrast to the cyclic desferrioxamine E, the simple lineal desferrioxamine B is formed by condensation of one unit of N^6^-acetyl-N^6^-hydroxycadaverine and two units of N^6^-succinyl-N^6^-hydroxyl-cadaverine (Figure 5). Other related cyclic desferrioxamines produced by several marine bacteria (nocardamines) are formed by a mechanism similar to that synthesizing desferrioxiamine E [120,121].

### 5.2. L-δ-N-Hydroxyornithine and D-δ-N-Formyl-N-Hydroxyornithine in the Biosynthesis of Bacterial Peptide Siderophores: Coelichelins

In addition to cadaverine-derived siderophores, there are bacterial peptide siderophores that arise from ornithine. Ornithine is synthesized from glutamate by a five-step pathway that forms part of the arginine biosynthetic route [122,123]. Another important bacterial siderophore is coelichelin, which is synthesized from ornithine-derived precursors. The coelichelin gene cluster, *cch*, was first identified in the genome of *S. coelicolor* [116], and an entirely similar cluster was found later in *Streptomyces ambofaciens* [33]. Deletion of the *cch* cluster prevents the formation of coelichelin in *S. coelicolor.* Moreover, expression of the *cch* gene cluster is regulated by iron starvation, and its control is exerted by the DmdR1/DmdR2 transcriptional iron regulators [124,125]. The precursors L-δ-N^5^-hydroxyornithine and D-δ-N^5^-formyl-N^5^-hydroxyornithine, together with D-threonine, form a peptide that constitutes the backbone of coelichelin, synthesized by an NRPS [31,33]. The three-modules protein, encoded by the *cchH* gene, contains three sites for binding the phosphopantetheine arm. The three amino acids activated by the NRPS are L-δ-N^5^-formyl-N^5^-hydroxyornithine, L-threonine, and L-δ-N^5^-hydroxyornithine. However, in the peptide, they appear in the D, D, L configuration, respectively. Modules 1 and 2 contain an epimerization domain that is known to be involved in the inversion of the amino acids’ stereochemical configuration. There is no thioesterase domain integrated into the CchH NRPS, but an additional gene, *cchJ*, encoding a thioesterase, is required for the release of the peptide formed. Two genes present in *S. coelicolor* and *S. ambofaciens cch* clusters, named *cchA* and *cchB*, encode, respectively, a monooxygenase and an acyltransferase that have been proposed to be responsible for the modifications of ornithine to its derivatives N^5^-Hydroxyornithine and N^5^-Formyl-N^5^Hydroxyornithine [33]. The monooxygenase encoded by *cchA* introduces a hydroxyl group at the C-5 amino group of ornithine, while the enzyme encoded by *cchB* formulates the NH_2_ group at the same position. Lautru and coworkers [31] purified coelichelin and determined its chemical structure. The experimentally determined structure of coelichelin corresponds to a tetrapeptide containing two molecules of threonine, and the authors claim that this tetrapeptide arises by iterative action of module 2 of the NRPS [31]. This abnormal product of a tri-modular NRPS, which does not adjust to the collinearity rule of the NRPS modules, has also been found in other peptide synthetases [126].

## 6. Siderophores Derived from Ornithine in Filamentous Fungi

Siderophores play a key role in the metabolism of filamentous fungi in soil and in infected plant and animal tissues [127,128]. In aerobic environments at neutral or alkaline pH, the ferrous ion is oxidized to ferric hydroxide, which is highly insoluble; therefore, soil microorganisms have difficulty obtaining sufficient iron in some habitats. Soil-dwelling filamentous fungi have developed several types of siderophores to scavenge iron. Moreover, some fungal pathogens, such as *A. fumigatus*, have a strong dependence on iron acquisition for their colonization of tissues and pathogenicity in humans [129,130,131].

From the biosynthetic point of view, at least two different classes of iron siderophores are formed in filamentous fungi: (1) siderophores formed by condensation of repeated units of N^5^-hydroxyornithine derivatives, and (2) peptide siderophores formed by NRPSs that include proteinogenic and ornithine-derived non-proteinogenic amino acid [132]. The iron is chelated in these siderophores by the formation of hydroxamates, although there are other chemical structures that also chelate iron. Three major types of hydroxamate-forming siderophores occur in fungi, namely fusarinins, ferrichromes, and coprogens [130,133]. Fungi contain high-affinity siderophores such as the fusarinine class, i.e., triacetylfusarinin, and low-affinity intracellular siderophores such as ferricrocin.

### 6.1. Fungal Siderophores Synthesized by Condensation of Acylated N^5^-Hydroxylornithine Units: Fusarinins

This siderophore class includes fusarinines and triacetylfusarinin (TAFC). The structure of fusarinine consists of three N^5^-anhydromevalonyl-N^5^-hydroxyornithine cyclically linked by ester bonds (Figure 6). The biosynthesis of this precursor starts with the hydroxylation of ornithine by the N^5^-ornithinehydroxylase encoded by the *sidA* gene [35,36,37], an enzyme that belongs to the N-hydroxylating flavoprotein monooxygenases [36]. The second half of the N^5^-anhydromevalonyl-N^5^-hydroxyornithine precursor starts with mevalonate, which is activated with CoA by SidI and then dehydrated by the SidH enzyme, respectively [34]. The anhydromevalonyl-CoA precursor transfers its anhydromevalonic moiety to N^5^-hydroxyornithine to form N^5^-anhydromevalonyl-N^5^-hydroxyornithine by a ligase encoded by *sidF* [36] and finally, three units of this intermediates are condensed to form fusarinine C by an NRPS-like protein encoded by the *sidD* gene. This NRPS is activated by phosphopantetheinylation by a PPT enzyme [134], as occurs with other NRPSs. Final acetylation by SidG using acetyl-CoA forms TAFC.

The three enzymes SidI, SidH, and SidF involved in the synthesis of N^5^-anhydromevalonyl-N^5^-hydroxyornithine, are peroxisomal enzymes as shown by green fluorescens protein targeting studies [135] and contain peroxisomal targeting sequences, either PST1 or PST2. The TAFC siderophores are extracellular in *A. fumigatus* and other filamentous fungi. Other extracellular secondary metabolites are known to be formed by enzymes located in peroxisomes (e.g., penicillin in *P. chrysogenum*), but the secretion mechanism of the siderophores or the β-lactam antibiotics from peroxisomes to the extracellular medium is still poorly known [136]. 

The extracellular TAFC siderophore binds to iron and transports it through the membrane to the cytosol, where iron is released by an esterase that disrupts the fusarinin structure.

Genes similar to *sidI* and *sidH* occur in many filamentous fungi. A cluster of genes *sidDFH* is present in *P. chrysogenum* (KZN90633 to KZN90636), as well as genes with high identity percentages to *sidA*, *sidI*, and *sidL*, suggesting that *P. chrysogenum* may form a siderophore of this class. Indeed, Kiel et al. [137] reported that the SidF protein is present in the peroxisomal matrix of *P. chrysogenum* [137].

### 6.2. Siderophores Synthesized by NRPSs in Filamentous Fungi: Ferrichromes

Ferrichromes are a family of cyclic hexapeptide siderophores. The biosynthesis of ferricrocin has been studied in *A. nidulans*, *A. fumigatus*, and *M. grisea* [138], and it seems that this ferrichrome occurs in many filamentous fungi. Each filamentous fungus contains one or two different ferrichromes differing in their peptide amino acids sequence, e.g., two different NRPSs were found to be involved in the biosynthesis of ferricrocin and the ferrichrome AS2488059, respectively, in *Acremonium persicinum* [139]. Ferricrocin is intracellular, whereas ASP2488059 is an extracellular siderophore that is proposed to serve as an iron-chelating carrier for iron transport. Similar ferrichromes with specific hexapeptides have also been found in the Basidiomycete *Ustilago maydis* [140].

The ferricrocin hexapeptide consists of three amino acids (serine and two molecules of glycine) and three molecules of the non-proteinogenic amino acid N^5^-acetyl-N^5^-hydroxyornithine [131]. The formation of ferricrocin shares the first step with the biosynthesis of TAFC (see above), e.g., the SidA enzyme forms N^5^-hydroxyornithine for both siderophore biosynthetic pathways [35]. A second step, specific for ferricrocin, is the acetylation of N^5^-hydroxyornithine using acetyl-CoA by two acetyltransferases; one of them constitutive, SidL, and another, iron-regulated acetyltransferase, still uncharacterized [38,135]. The assembly of serine, two molecules of glycine, and three molecules of N^5^-acetyl-N^5^-hydroxyornithine to form the ferricrocin hexapeptide is catalyzed by the NRPS encoded by *sidC.* This NRPS requires to be modified by the same PPT described above for fusarinins [134]. The sequence of modules of the NRPS determines the identity of each peptide siderophore formed.

### 6.3. Coprogens

Coprogens were first reported in *Neurospora crassa* but later were studied in detail in the citrus plant pathogen *Alternaria alternata* [133]. Several *Alternaria* species produce the dimethylcoprogen siderophore [141].

The simplest siderophore of this class is rhodotorulic acid produced by species of the yeast-basidiomycete *Rhodotorula.* This compound is a dipeptide consisting of two units of N^5^-acetyl-N^5^-hydroxylornithine, forming a diketopiperazine ring by head-to-head condensation. Rhodotorulic acid is the core moiety of coprogens. Other coprogens contain, in addition, a third N^5^-acetyl-N^5^-hydroxy-ornithine molecule attached to the diketopiperazine ring by an ester bond [39]; following its assembly, the compound is secreted to the extracellular medium. More complex coprogens have been described in highly pathogenic *Alternaria longipes*, such as neocoprogen I and isoneocoprogen I [39].

An NRPS named NPS6 was found to be involved in the condensation of the three N^5^-acyl-N^5^-hydroxyornithine units. Deletion of the *nps6* gene in *Alternaria brassicicola*, a plant pathogen, showed that the coprogen synthesized by *nps6* is a pathogenic determinant and also is involved in oxidative stress resistance [141]. Later, Chen et al. [142] obtained a mutant of the citrus pathogen *A. alternata* defective in the Nps6 protein and found that this mutant was unable to scavenge iron from citrus plants and was non-pathogenic. Complementation of this mutant with the *nps6* gene resulted in a partial restoration of the siderophore activity, suggesting that Nps6 is responsible for coprogen biosynthesis. Expression of the *nps6* gene is regulated by the iron concentration in the culture medium and indicates that the *A. alternata* pathogenicity is influenced by the iron availability in the soil and the citrus plant. Genes orthologous to *nps6* are present in many other fungi, some of them plant pathogens, such as *Cochliobolus heterostrophus* and *Fusarium graminearum* [141,143].

## 7. Role of Intracellular and Extracellular Siderophores

Both intracellular and extracellular siderophores are found in many filamentous fungi [144]. In contrast to animals, plants, and some bacteria that store iron complexed to the protein ferritin, filamentous fungi lack ferritin. Instead, filamentous fungi use some intracellular siderophores for cellular iron storage. However, these intracellular siderophores need to be compartmentalized, e.g., in vacuoles or biochemically protected, to avoid toxicity due to their iron chelating activity. Ferricrocin has been identified to serve as an intracellular siderophore in several fungi, including *A. nidulans*, *F. graminearum*, and *M. grisea* [36,141]. In addition, in *A. fumigatus*, a distinct ferricrocin derivative, hydroxyferricrocin, is used for this purpose. The hydroxyferricrocin is synthesized from ferricrocin by a specific hydroxylase [36]. Inactivation of the *sidC* gene encoding the NRPS that forms ferricrocin and hydroxyferricrocin showed that these intracellular siderophores are required for germ tube formation, oxidative stress resistance, sporulation, and virulence. The biosynthesis of intracellular ferricrocin constitutes an interesting target for screening antifungal antibiotics since humans lack this siderophore. Summing up, siderophores are excellent targets for combating pathogenic microbial infections since antimicrobial agents that prevent the formation of siderophores will inhibit the growth of pathogens due to their requirement for iron.

## 8. Summary and Future Outlook

In the last decades, great advances have been made in the characterization of the genomes and proteomes of microorganisms [145,146]. This has provided new information on the metabolic complexity of networks that provide precursors for the biosynthesis of various secondary metabolites. We have discussed in this article the present knowledge on the biosynthetic pathways that convert amino acids, such as lysine and ornithine, into a wealth of metabolites that have important applications in medicine and the pharmaceutical industry, including antibacterial compounds, antifungals, immunosuppressants, antitumoral, neuroprotectives, and siderophore molecules. Remarkably, lysine is synthesized in different organisms, such as bacteria, fungi, and plants, by three distinct pathways, and this provides a large number of intermediates. An interesting network includes the biosynthesis of the α-AAA, α-aminoadipate semialdehyde, saccharopine, and some pathway derivatives such as pipecolic acid; there is a clear interconversion of precursors and intermediates that supply molecules for the biosynthesis of β-lactam antibiotics and many pipecolic-derived bacterial and plant metabolites. In *P. chrysogenum/P. rubens*, which serve as model microorganisms for the biosynthesis of β-lactams there are up to five genes coding for proteins similar to saccharopine reductase and saccharopine dehydrogenase that are likely to increase the supply of key precursors for this class of antibiotics; the presence of several copies of these genes, two of which are located inside or close to the penicillin gene cluster, indicates that evolutionary gene duplication and adaptation occurred in nature that may influence the survival of penicillin producing strains in competition with bacteria in the soil. Enzymes involved in the formation of α-aminoadipate semialdehyde and α-AAA have been developed both in β-lactam producer filamentous fungi and in actinobacteria. Noteworthy, the lysine aminotransferase in actinobacteria is encoded by a gene of the cephamycin gene cluster, co-regulated with other genes of the cluster, suggesting that there is a convergent adaptation of enzymes in different groups of microorganisms to provide the above-mentioned precursors. Another important group of precursors derived from the lysine or ornithine pathways is essential for the production of bacterial and fungal siderophores. This article provides evidence indicating that there are important contributions to the formation of lysine derivatives, such as cadaverine, or ornithine derivates, such as N^5^-hydroxyornithine or N^5^-succinyl-N^5^-hydroxyornithine, for the biosynthesis of different types of bacterial siderophores, such as desferrioxamines and coelichelin. In addition, some fungal siderophores, such as fusarinins, ferrichromes, and coprogens, derive from ornithine. In this review, we have integrated all enzymatic studies with the characterization of the genes that encode those enzymes. Heterologous expression of genes encoding the biosynthetic precursors is a very interesting possibility to improve the production of bioactive compounds. So far, expression of core genes of the β-lactam antibiotic has been performed in other fungi and yeasts [147,148], but there are problems derived from the proper cellular localization of the cloned biosynthetic enzymes and the intracellular transport of intermediates through organelles [136]. Similarly, the cephamycin genes of *S. clavuligerus*, including *lat* and *pcd*, have been expressed in the model organism *S. coelicolor* [149] and other *Streptomyces* species, but the final antibiotic production is lower than in *S. clavuligerus.* This finding may be due to a deficit in the supply of precursors or to the inadequate expression of some of the biosynthetic genes for transcriptional factors in the host strain. An important conclusion is that the networks that provide these precursors are more complex than it was previously believed, and future developments will contribute to increasing the availability of these precursors for bioactive compounds. Progress in this field will contribute to advancing the field of synthetic biology to produce semisynthetic and chemically synthesized novel biological products.

## Figures and Tables

**Figure 1 antibiotics-12-00159-f001:**
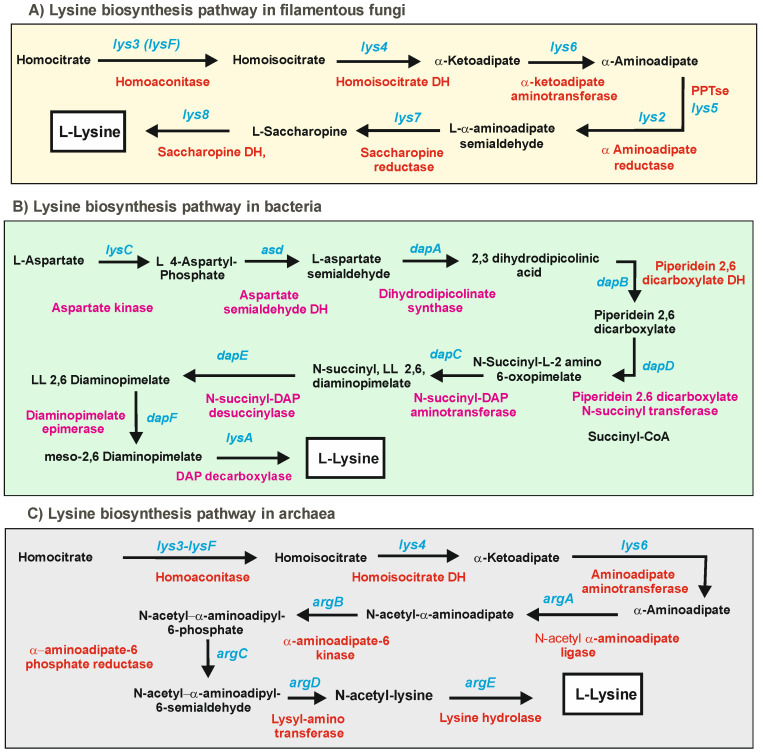
Biosynthetic pathways of lysine. (**A**) In filamentous fungi. (**B**) In bacteria. (**C**) In *Thermus thermophilus* and archaea. The enzymes involved in each step are indicated in red letters. The genes are indicated in blue letters. Note: For simplicity, the designation of the genes of yeasts and filamentous fungi are used with the three letters in cursive letters in the text. The designation of the genes in all figures refers to the filamentous fungi names published in the literature (not to the yeast names, which are frequently different).

**Figure 2 antibiotics-12-00159-f002:**
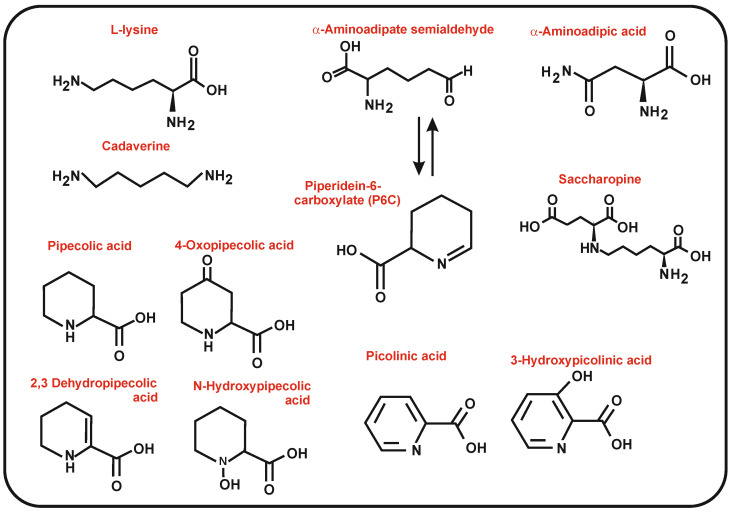
Chemical structure of intermediates of the lysine biosynthesis pathway (lysine, saccharopine, α-Aminoadipate semialdehyde, Piperideine-6-Carboxilic acid, and α-Aminoadipic acid) or compounds derived from lysine (pipecolic acid, 4-Oxo-pipecolic acid, 2-3 Dehydropipecolic acid, N-hydroxypipecolic acid, and 3 Hydroxypicolinic acid) that are precursors of secondary metabolites. The interconversion of α-aminoadipate semialdehyde in P6C is shown by a double arrow.

**Figure 3 antibiotics-12-00159-f003:**
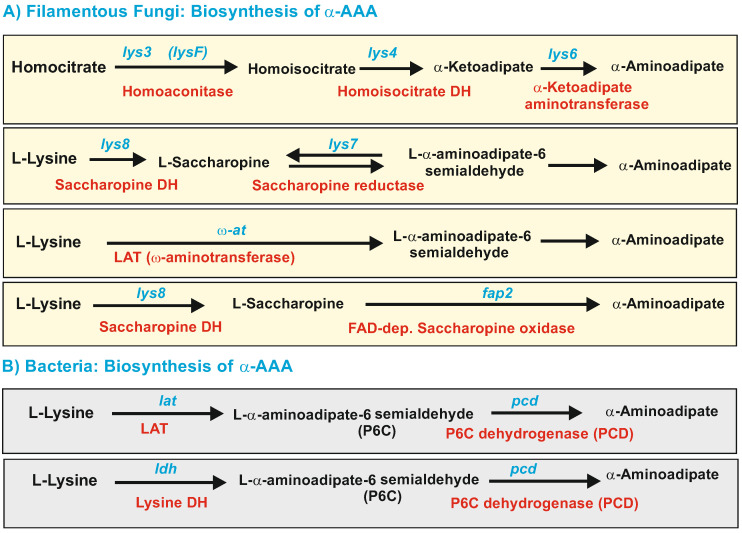
Biosynthetic pathways of α-Aminoadipic acid. (**A**) Biosynthesis of α-Aminoadipic acid in filamentous fungi. All the pathways have a yellow background. (**B**) Biosynthesis of α-Aminoadipic acid in bacteria. All the pathways have a gray background. The enzymes involved in every step are shown in red letters; the genes are shown in blue letters. Note that the enzyme forming α-AAA from aminoadipate-6 semialdehyde in fungi is unclear (see text); the name of the gene encoding the lysine dehydrogenase of *A. tumefaciens* has been designated *ldh*.

**Figure 4 antibiotics-12-00159-f004:**
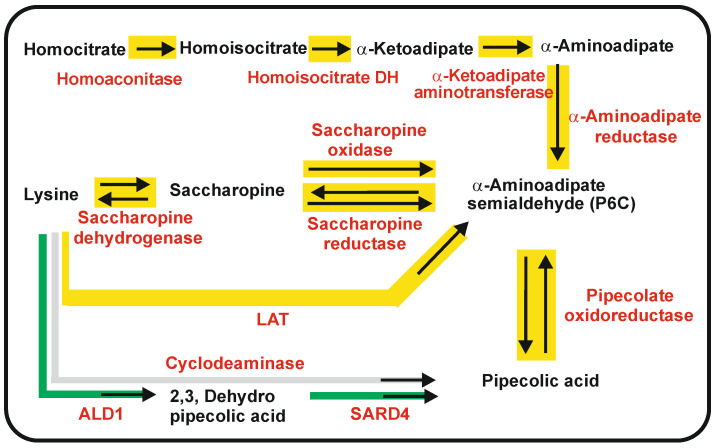
Biosynthetic pathways converging in the formation of pipecolic acid. The biosynthesis pathways of pipecolic acid in filamentous fungi are drawn on a yellow background. The biosynthesis pathways of pipecolic acid in bacteria are drawn on a gray background. The biosynthesis pathway of pipecolic acid in plants is drawn on a green background. Precursors, intermediates and final products of the pathways are in black letters. The enzymes are indicated in red letters. Note that pipecolic acid may be formed in a biosynthetic pathway from α-aminoadipate or from catabolism of lysine.

**Figure 5 antibiotics-12-00159-f005:**
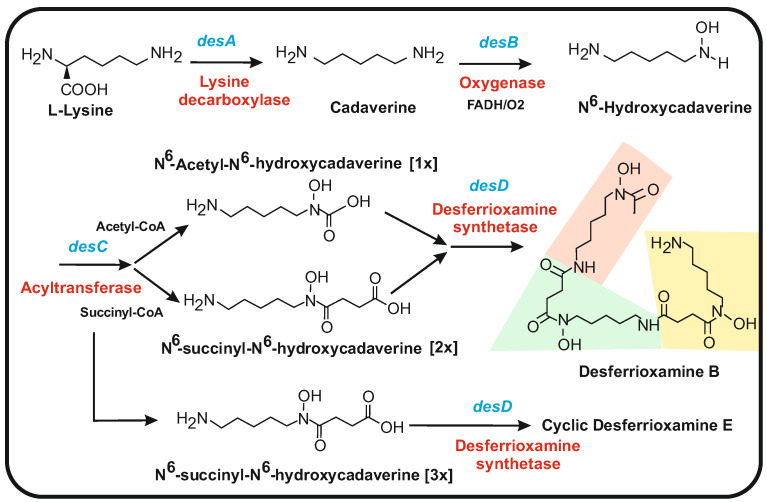
Biosynthetic pathway of desferrioxamines. The biosynthetic pathways of desferrioxamines B and E are shown. The enzymes responsible of every step are indicated in red letters. The encoding genes are shown in blue letters. The structure of the lineal desferrioxamine B is shown, indicating the position of the N^6^-acetyl-N^6^ Hydroxycadaverine (pink) and of the two molecules of N^6^-succinyl-N^6^-Hydroxycadaverine (yellow and green colors).

**Figure 6 antibiotics-12-00159-f006:**
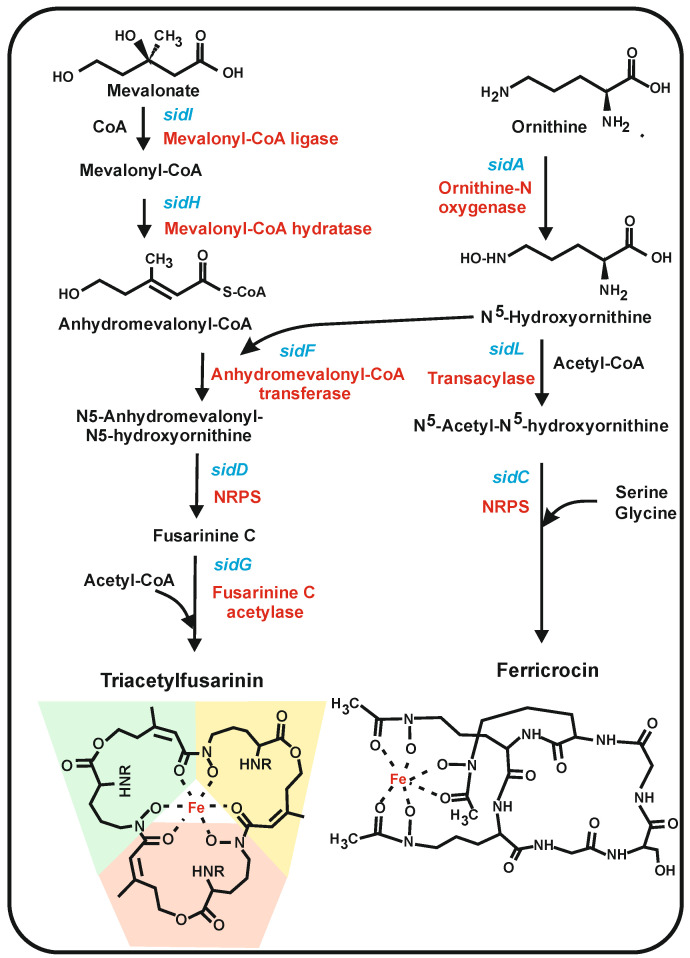
Biosynthesis of the siderophores triacetylfusarinine and ferricrocin. Both biosynthetic pathway and their interaction are shown. The enzymes are labeled in red letters; the genes are indicated in blue letters. At the bottom of the figure are shown the structures of triacetylfusarinine and ferricrocin and the position of the chelated iron atom forming hydroxamates. The three molecules of acetylated N^5^-anhydromevalonyl-N^5^-hydroxyornithine that form the TAFS structure are shadowed in yellow, green, and pink, respectively.

**Table 1 antibiotics-12-00159-t001:** Compounds derived from lysine or ornithine that are precursors of secondary metabolites.

	SM Precursor	Precursor Synthesizing Enzyme(s)	Secondary Metabolite (SM)	Producer Strain	Reference
1	α-Aminoadipic acid (α-AAA)	α-Ketoadipate aminotransferase	Penicillins Cephalosporin C Cephamycins	*Penicillium chrysogenum* *Acremonium chrysogenum* *Streptomyces clavuligerus*	[5,18]
2	α-Aminoadipic semialdehyde (P6C)	α-AAA reductase Lysine-6-aminotransferase	Penicillins, Cephalosporin C Cephamycins	*P. chrysogenum* *A. chrysogenum* *S. clavuligerus*	[19,20,21]
3	Saccharopine	Saccharopine reductase and Pipecolate oxidoreductase	Pipecolic acid	Most fungi	[22,23,24]
4	4-oxo-pipecolic acid	Pipecolate monoxygenase P_450_	Swainsonine, Slaframine	*Slafractonia leguminicola* *Metarhizium anisopliae* *Undifilum oxytropis*	[5]
5	Pipecolic acid	Lysine cyclodeaminase	Rapamycin Pristinamycin I Tacrolimus	*Streptomyces hygroscopicus* *Streptomyces pristinaespiralis* *Streptomyces tsukubaensis*	[25,26,27]
6	3-Hydroxypicolinic acid	Lysine-2-aminotransferase with dehydrase activity	Pristinamycin I Virginiamycin	*S. pristinaespiralis* *Streptomyces virginiae*	[28,29]
7	Cadaverine	Lysine decarboxylase	Desferrioxamines	*Streptomyces coelicolor* *Streptomyces pilosus*	[30]
8	N^5^-Hydroxyornithine N^5^-Formyl-N^5^-Hydroxy- ornithine	N^5^-ornithine monooxygenase N^5^-hydroxyornithine acyltransferase	Coelichelin	*S. coelicolor* *Streptomyces ambofaciens*	[31,32,33]
9	N^5^-anhydromevalonyl-N^5^-hydroxyornithine	N^5^-ornithine monooxygenase Mevalonil CoA ligase Mevalonyl-CoA dehydrase	Fusarinine, TAFC	Many fungi	[34,35]
10	N^5^-acetyl-N^5^-hydroxyornithine	N^5^-ornithine monooxygenase N^5^-hydroxyornithine acyltransferase	Ferrichrocin	Many fungi	[35,36,37,38]
11	N^5^-acetyl-N^5^-hydroxyornithine	N^5^-ornithine monooxygenase N^5^-hydroxyornithine acyltransferase	Coprogen Methyl-coprogen	Many fungi *Alternaria alternata*	[39]

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
