# Peer review of "Interconnected Set of Enzymes Provide Lysine Biosynthetic Intermediates and Ornithine Derivatives as Key Precursors for the Biosynthesis of Bioactive Secondary Metabolites"

_antibiotics, 2023, doi:10.3390/antibiotics12010159_

Round 1
Reviewer 1 Report
1. Limit the key words to 6-5 (only key terms try to avoid abbreviations).
2. Remove the underline from the legend of figure3
3. In line 50, mention the different tissues.
4. In line 96, change investigate to investigated.
5. Add the genes involved in Biosynthetic pathways of -Aminoadipic acid.
6. In figure 4, mention the enzyme involved in conversion of Homocitrate to -Aminoadipate.
7. Format the references according to the journal and avoid unwanted spacing.
8. Give the list of genes/enzymes involved in Pipecolic acid biosynthesis (both plants and microbes) in a table.
9. List the genes involved in biosynthesis of 4-oxopipecolic acid and 3-hydroxypicolinic acid in a table.
10. The suggested tables can be added as a supplementary data
Author Response
Reviewer1. Comments and Suggestions for Authors
- Limit the key words to 6-5 (only key terms try to avoid abbreviations).
Answer: The number of keywords has been reduced to the minimal possible and the abbreviations have been eliminated.
- Remove the underline from the legend of figure3
A: Changed as suggested in all figures.
- In line 50, mention the different tissues.
A: The paragraph has been modified according to the published articles as follows “…and a-aminoadipate semialdehyde in bovine and primate liver and human placenta” (Line 48)
- In line 96, change investigateto investigated.
A: Changed as suggested
- Add the genes involved in Biosynthetic pathways of a-Aminoadipic acid.
A: the name for the genes involved in AAA biosynthesis are shown in Fig 3. These are the name of the genes in filamentous fungi. Note that these genes names might be different from those of yeasts; this has been indicated in the legend of Figure 1. In the legend of Fig 4 we have added “the genes are shown in blue letters”. The name for the gene encoding the lysine dehydrogenase from Agrobacterium tumefaciens has been named ldh according to the enzyme designation (Misono et al., 1989) and has been added to legend of the figure 3.
- In figure 4, mention the enzyme involved in conversion of Homocitrate to a-Aminoadipate.
A: The name of the enzymes converting Homocitrate to a-Aminoadipate have been included in Fig 4 as requested
- Format the references according to the journal and avoid unwanted spacing.
A: The spaces have been removed. As far as we know the format of the references is OK. We checked the references and found some minor errors. However, we will be happy to make any additional corrections if you indicated it.
- Give the list of genes/enzymes involved in Pipecolic acid biosynthesis (both plants and microbes) in a table.
A: Table S1 has been included as supplementary material and a sentence has been added to the text indicating “The enzymes and genes involved are shown in Table S1” (Line 314).
- List the genes involved in biosynthesis of 4-oxopipecolic acid and 3-hydroxypicolinic acid in a table.
A: Table S2 has been included as supplementary material, and the following sentence has been included in the text “The enzymes and genes involved in the biosynthesis of these molecules are shown in Table S2” (Line 400).
Reviewer 2 Report
The review article entitled “Interconnected set of enzymes provide lysine biosynthetic intermediates and ornithine derivatives as key precursors for the biosynthesis of bioactive secondary metabolites” is well written and the similarity index with the literature is also found to be low (16%). However, it requires minor revision on the following points.
1) In the abstract, ‘phylogenetic relationships between the genes encoding the enzymes’ were planned but there is no such report in the manuscript. It would be interesting to readers such phylogenetic relationships.
2) The manuscript is descriptive in other areas rather than sequence analyses of genes involved in the biosynthesis of amino acids and derivatives.
3) In figure 1, it is suggested to show the chemical structures of molecules or intermediates involved in lysin biosynthesis. And report the genes (proteins) involved in the biosynthesis as per GenBank or PDB.
4) In figure 3, please draw chemical structures and here as well report the genes (proteins) involved in the biosynthesis as per GenBank or PDB. Did you see evidence of horizontal gene transfer between fungus and bacteria in the case of α-aminoadipate biosynthesis?
5) In other areas of the manuscript as well, please compare proteins of similar functions and give your opinion on the expression of those gene clusters in heterologous host.
Author Response
The review article entitled “Interconnected set of enzymes provide lysine biosynthetic intermediates and ornithine derivatives as key precursors for the biosynthesis of bioactive secondary metabolites” is well written and the similarity index with the literature is also found to be low (16%). However, it requires minor revision on the following points.
- In the abstract, ‘phylogenetic relationships between the genes encoding the enzymes’ were planned but there is no such report in the manuscript. It would be interesting to readers such phylogenetic relationships.
Answer: We have performed analysis of functional relationship between biosynthetic enzymes of different bioactive metabolites pathways but a formal phylogenetic analysis is out of the scope of this article. Phylogenetic analysis requires some additional experimental work that we will perform in future articles. Therefore, we have modified the final sentence of the abstract indicating this fact.
- The manuscript is descriptive in other areas rather than sequence analyses of genes involved in the biosynthesis of amino acids and derivatives.
Answer: Yes, we have carried out a detailed analysis of the genes and the encoded proteins which are involved in the biosynthesis of precursors of bioactive metabolites (see sections 3, 4, 7 and 8). The content is based on the available scientific information but obviously further experimental research is required in several of these examples. At present time the article compares all the recent available published research work.
- In figure 1, it is suggested to show the chemical structures of molecules or intermediates involved in lysin biosynthesis. And report the genes (proteins) involved in the biosynthesis as per GenBank or PDB.
Answer: The most of the important structures can be seen in Fig 2. We have tried to draw a figure including all the chemical intermediates and final products as well as the proteins involved. This is too much information and cannot be seen, because the structures take up too much space and the lettering cannot be read. The gene names have been included; these names correspond to those of filamentous fungi (that might be different than those of yeasts); this is indicated in the legend of Figure 1.
- In figure 3, please draw chemical structures and here as well report the genes (proteins) involved in the biosynthesis as per GenBank or PDB. Did you see evidence of horizontal gene transfer between fungus and bacteria in the case of α-aminoadipate biosynthesis?
A: We have tried to draw a figure including both the structures of the intermediates and the final products and the enzymes involved in each step. As indicated above the figure becomes too large and it is impossible to read the letters. We have included the names of the genes involved in these pathways in filamentous fungi as described in the scientific literature (Naranjo et al, 2001; Bañuelos et al., 2000; Casqueiro et al 1999 and in the Databases). In relation to the horizontal transfer of the lat gene (lines 201-203) we describe that the lysine aminotransferase of filamentous fungi (Naranjo et al 2001) is related to the bacterial lysine aminotransferase characterized in several cephamycin producing actinobacteria but although they have the same function the amino acids identity is low (27% identity, 42% similarity). Therefore, it does not seem to have been transferred by HGT. At difference of the core genes of the isopenicillin N biosynthesis (Ref. 56). In conclusion the lat genes of fungi and bacteria appear to have evolved convergently.
- In other areas of the manuscript as well, please compare proteins of similar functions and give your opinion on the expression of those gene clusters in heterologous host.
A: In the text we refer to some examples of heterologous expression, e.g. the heterology expression of the lat gene (ref 86, Fuji et al) but we have added a paragraph (Lines 672-682) in Future Outlouk to describe the available information on the heterologous expression of genes involved b-lactam biosynthesis in yeast and Aspergillus species (Awan et al., 2017; Smith et al 1990) and in other Streptomyces species different from the producer strain (Martinez-Burgo et al., 2014)